

# The relationship between PLOD1 expression level and glioma prognosis investigated using public databases

Lei Tian[1], Huandi Zhou[1], Guohui Wang[1], Wen yan Wang[1], Yuehong Li[2] and Xiaoying Xue[1]

[1] Department of Radiotherapy, The Second Hospital of Hebei Medical University, Shijiazhuang, Hebei, China
[2] Department of Pathology, The Second Hospital of Hebei Medical University, Shijiazhuang, Hebei, China

Corresponding author
Xiaoying Xue, xxy0636@163.com

## ABSTRACT

**Background**. Glioma is the most common type of intracranial tumor with high malignancy and poor prognosis despite the use of various aggressive treatments. Targeted therapy and immunotherapy are not effective and new biomarkers need to be explored. Some Procollagen-lysine 2-oxyglutarate 5-dioxygenase (*PLOD*) family members have been found to be involved in the metastasis and progression of tumors. Both *PLOD2* and *PLOD3* had been reported to be highly expressed in gliomas, while the prognostic value of *PLOD1* remains to be further illustrated, so we want to investigate the *PLOD1* expression in glioma and its clinical implication.

**Methods**. We collected gene expression and corresponding clinical data of glioma from the Chinese Glioma Genome Atlas (CGGA) database, The Cancer Genome Atlas (TCGA) database and the Gene Expression Omnibus (GEO) database. First, we analyzed the expression and mutation of *PLOD1* in gliomas and its relationship with clinicopathologic characteristics. Then, we conducted survival analysis, prognostic analysis and nomogram construction of the *PLOD1* gene. Finally, we conducted gene ontology (GO) enrichment analysis and gene set enrichment analysis (GSEA) to explore possible mechanisms and gene co-expression analysis was also be performed.

**Results**. The results showed that the expression level of *PLOD1* was higher in gliomas than normal tissues, and high expression of *PLOD1* was related to poor survival which can serve as an oncogenic factor and an independent prognostic indicator for glioma patients. Both the GO and GSEA analysis showed high expression of *PLOD1* were enriched in Extracellular matrix (ECM) related pathways, the co-expression analysis revealed that *PLOD1* was positively related to *HSPG2*, *COL6A2*, *COL4A2*, *FN1*, *COL1A1*, *COL4A1*, *CD44*, *COL3A1*, *COL1A2* and *SPP1*, and high expression of these genes were also correlated to poor prognosis of glioma.

**Conclusions**. The results showed that high expression of *PLOD1* leads to poor prognosis, and *PLOD1* is an independent prognostic factor and a novel biomarker for the treatment of glioma. Furthermore, targeting *PLOD1* is most likely a potential therapeutic strategy for glioma patients.

## INTRODUCTION

Gliomas are the most common type of primary intracranial tumor with high malignancy and poor prognosis, especially in high-grade gliomas (*Ostrom et al., 2020*). The current standard approach of treatment is maximum surgical resection followed by adjuvant radiotherapy and chemotherapy (*Van den Bent et al., 2017*; *Stupp et al., 2009*), and newly developed alternating electric field therapy has been recommended for glioblastoma (GBM) in recent years (*Sampson, 2015*). Despite the using of multiple aggressive therapies, the overall survival of gliomas remained poor, so it is urgent to explore more effective treatment methods. In recent years, immunotherapy and targeted therapy have made great progress in many other tumors, but have little effect in glioma (*Fecci & Sampson, 2019*), and although some of the molecular features showed some prognostic values (*Cairncross et al., 2012*; *Hegi et al., 2005*), none of them had became a target for targeting therapy, thus developping novel and targeted therapeutic options is urgent.

Extracellular matrix (ECM) is an important constituent of tumor microenvironment, corelating with tumor development and progression. Among the various ECM components, collagens are the most abundant proteins, and its deposition and cross-linking are closely related to tumor proliferation and invasion (*Jover et al., 2018*). Procollagen-lysine 2-oxyglutarate 5-dioxygenase (*PLOD*) catalyzed hydroxylysine residue, which is critical for the formation of covalent cross-link (*Qi & Xu, 2018*). An increasing number of evidences indicate that the *PLOD* family, which consists of *PLOD1*, *PLOD2* and *PLOD3*, plays an important role in the development and progression of tumors. Both *PLOD2* and *PLOD3* had been reported to be highly expressed in gliomas and were associated with tumor progression and prognosis (*Song et al., 2017*; *Tsai et al., 2018*). Some previous studies revealed that *PLOD1* promoted tumorigenesis and metastasis in osteosarcoma, bladder cancer and esophageal squamous cell carcinoma (*Wu et al., 2020*; *Yamada et al., 2019*; *Li et al., 2017*), while the expression and prognostic role of *PLOD1* in glioma remain to be further illustrated.

Bioinformatics analysis using high-throughput sequencing and clinical data is developing rapidly to identify sensitive biomarkers and prognostic factors for a variety of tumors, including gliomas. The acetylation modification and kinase activity of *PAK1* were considered to be an instrumental role in hypoxia-induced autophagy initiation and maintaining GBM growth, and *PAK1* might represent potential therapeutic targets for GBM treatment (*Feng et al., 2020*). *HIST1H2BK* was identified as an indicator of poor prognosis and a promising biomarker for the treatment of low-grade glioma (LGG) (*Liu et al., 2020*). Similarly, *ARL9* had been shown its prognostic value in LGG, and probably played an important role in immune cell infiltration (*Tan et al., 2020*). All these markers are possible used for advanced decision-making processes in the future; however, many more potential prognostic indicators still need to be explored in gliomas.

Therefore, this study attempts to analyze the *PLOD1* gene expression levels in glioma and normal tissue, using public database. We also explored the relationship between *PLOD1* expression and clinical characteristics as well as prognosis. Finally, we identified the *PLOD1*-related signaling pathways and suggested that *PLOD1* acted as a cancer-promoting

factor in tumor progression, providing a potential prognostic biomarker and therapeutic target for glioma patients.

## MATERIALS & METHODS

### Data collection and download

The clinical data and gene expression data were obtained from Chinese Glioma Genome Atlas (CGGA, http://www.cgga.org.cn/) database. Two datasets were downloaded containing 1018 samples (*Wang et al., 2015*; *Liu et al., 2018*; *Bao et al., 2014*; *Zhao et al., 2017*) up to May 6, 2020, and 20 non-glioma brain tissues were downloaded for analyzed. In addition, The Cancer Genome Atlas (TCGA, https://portal.gdc.cancer.gov/) database containing 592 glioma samples was collected for validation. There were 449 LGG samples and 143 GBM samples, respectively. Then, we searched "glioma" and "GEO" in the Gene Expression Omnibus (GEO, https://www.ncbi.nlm.nih.gov/geo/) database and selected GSE4290, GSE7696 and GSE50161 as validation sets. The GSE4290 dataset contains 153 glioma samples and 23 normal samples (*Sun et al., 2006*), the GSE7696 dataset contains 80 glioma samples and 4 normal samples (*Lambiv et al., 2011*), and the GSE50161 dataset contains 117 glioma samples and 13 normal samples (*Griesinger et al., 2013*). Before further analysis, RNA sequencing data were log2-transformed. All these databases were screened to eliminate samples missing clinical information.

### Gene expression analysis, GEPIA database analysis, mutation analysis and Cancer Cell Line Encyclopedia (CCLE) analysis

The expression data of *PLOD1* in normal brain tissue and glioma in CGGA dataset were imported into GraphPad Prism 8 software for analysis, and then validated by the GSE4290 dataset. In addition, *PLOD1* expression analyses of GBM and LGG were also performed in GEPIA (http://gepia.cancer-pku.cn/) (*Tang et al., 2017*). In addition, we performed mutation analysis to better comprehend the genomics profile of *PLOD1* based on the cBioPortal online database (*Gao et al., 2013*). Furthermore, we used the CCLE database (https://portals.broadinstitute.org/ccle/home) to assess *PLOD1* expression in different cancers.

### Correlations between *PLOD1* expression and clinical outcomes and clinicopathologic characteristics

The data from the CGGA datasets were mainly used to research the prognostic role of *PLOD1* in gliomas by using R software. According to the median expression level of *PLOD1*, high and low group were divided. The "survival" and "survminer" package were used in R software to plot survival curves for different *PLOD1* expression level. And then, the survival data of different IDH and MGMT promoter methylation status were imported into GraphPad Prism 8 software for survival analysis, so as to explore the survival of *PLOD1* expression levels in different molecular types of gliomas. We used the "survival ROC" package to calculate receiver operator characteristic (ROC) curves for *PLOD1* at 1, 3, and 5 years using the Kaplan–Meier method. Univariate and multivariate Cox analysis were also performed to assess the predictive value of *PLOD1* at a significance level of $P < 0.001$.

Based on the TCGA datasets, survival curve and ROC curves were performed to validate. To develop an individual prognostic signature for the 1-, 2- and 3-year survival rates, we constructed a nomogram in CGGA cohort using the "survival" and the "rms" package in R software. Following that, calibration curves were plotted to evaluate the concordance between actual and predicted survival. In addition, the correlationship between *PLOD1* expression and clinicopathologic characteristics was performed using the "beeswarm" package in R software.

## Differential genes expression analysis and enrichment analysis for significant pathways

The mRNA sequencing datas in glioma from CGGA datasets were normalized and the differentially expressed genes (DEGs) including significantly upregulated and downregulated genes were screened with an adjusted *p* value <0.05 and absolute log2 fold change (FC) >2, and then a volcano plot of DEGs was generated using the "limma" package in R software (*Ritchie et al., 2015*). Using the screened DEGs, gene ontology (GO) enrichment analysis was performed on the online tool-Metascape (*Zhou et al., 2019*) (http://metascape.org/gp/index.html#/main/step1). In addition, gene set enrichment analysis (GSEA) was also performed to indirectly explain the function of *PLOD1* (*Subramanian et al., 2005*). A gene set was considered as an enriched group when the NES>1 and FDR score < 0.05.

## Co-expression analysis

The GO and KEGG analysis both enriched in extracellular matrix (ECM). The common genes of DEGs and the key genes in the ECM pathways enriched by the KEGG analysis were performed for correlation analysis between *PLOD1*. Pearson correlation analysis was used for parametric tests, Spearman correlation analysis was used for non-parametric tests. A circular plot and a pheatmap of the common genes positively associated with *PLOD1* were generated by R software. In addition, every common gene was preformed in GEPIA to examination the expression and survival in gliomas.

## Statistical analysis

Statistical analyses were performed with R software v3.6.3 (http://www.r-project.org/) (*Hjalt, Amendt & Murray, 2001*), and Prism 8 (GraphPad Software, Inc). The "survival" package (https://CRAN.R-project.org/package=survival), "survminer" package (https://CRAN.R-project.org/package=survminer), "survivalROC" package (https://CRAN.R-project.org/package=survivalROC), "rms" package, "beeswarm" package (https://CRAN.R-project.org/package=beeswarm), "limma" package, "ggplot2"package, "pheatmap" package (https://CRAN.R-project.org/package=pheatmap), "corrplot" package (https://github.com/taiyun/corrplot), and "circlize" package (*Gu et al., 2014*) of R software were used successively. Data were considered significant at $P < 0.05$.

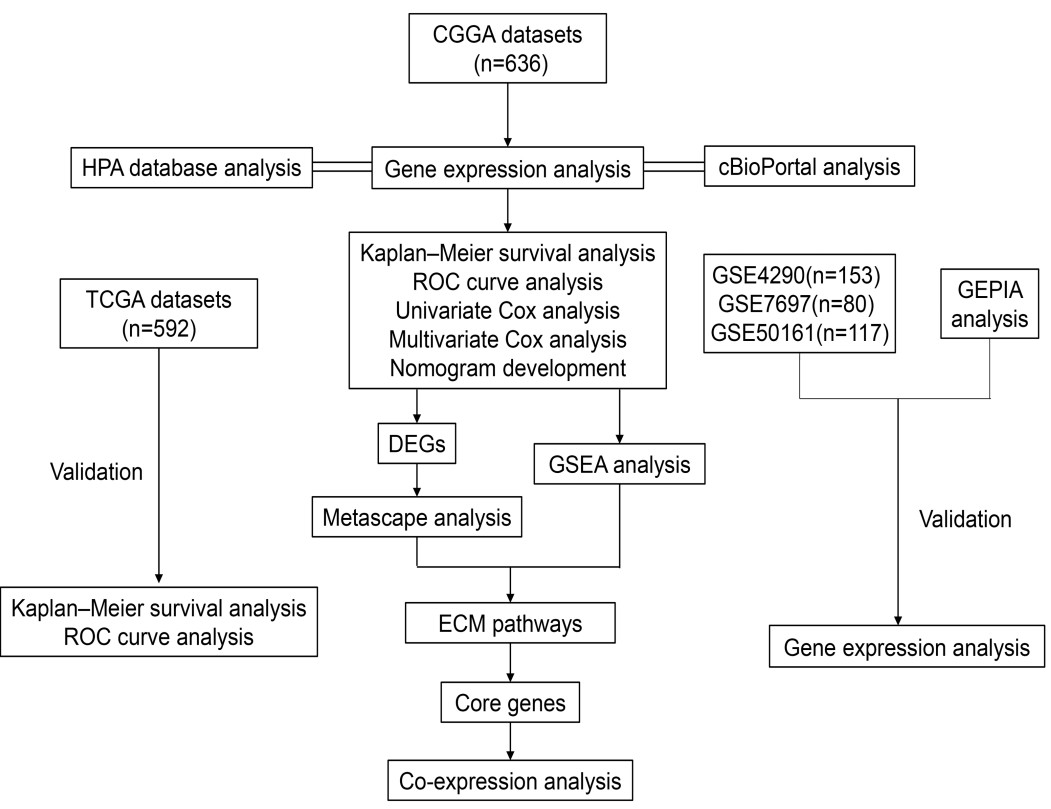

**Figure 1 Workflow of the whole study.** Abbreviations: CGGA, Chinese Glioma Genome Atlas; TCGA, The Cancer Genome Atlas; GSE, GEO Series; GEO, Gene Expression Omnibus; HPA, *The Human Protein Atlas; ROC,* Receiver Operator Characteristic; DEGs, Differentially Expressed Genes; GSEA, Gene Set Enrichment Analysis; ECM, Extracellular Matrix.

# RESULTS

## Characteristics of the samples

The workflow of our study is shown in Fig. 1. After screening, a total of 636, 592, 153, 80 and 117 glioma patient samples were obtained from the CGGA, TCGA, GSE4290, GSE7696 and GSE50161 datasets. Both CGGA and TCGA datasets contained grading data, age, gender, IDH mutation status and 1p19 codeletion status. Additionally, postoperative radiotherapy or chemotherapy follow-up and MGMT methylation data were only included in the CGGA dataset. Detailed clinical information classification and percentages of glioma patients are shown in Table 1.

## *PLOD1* gene expression and genomic characteristics in glioma

The expression level of *PLOD1* was significantly higher in gliomas than in normal tissues based on the CGGA and GEO datasets (Figs. 2A, 2B, 2C, 2D), and the same results were obtained in GEPIA online analysis in both GBM and LGG (Fig. 2E). The genomic alteration of *PLOD1* was shown in Fig. 2F, and several mutation types were provided using cBioPortal online database (Fig. 2G). In addition, the results from the CCLE database showed that

**Table 1  The characteristics of the public database samples.**

| Characteristics | CGGA ($n = 636$) | | TCGA ($n = 592$) | |
|---|---|---|---|---|
| | Case | Proportion | Case | Proportion |
| WHO Grade | | | | |
| II | 164 | 25.8% | 211 | 35.6% |
| III | 207 | 32.5% | 238 | 40.2% |
| IV | 265 | 41.7% | 143 | 24.2% |
| Age(years) | | | | |
| ≥42 | 342 | 53.8% | 349 | 59.0% |
| <42 | 294 | 46.2% | 243 | 41.0% |
| Gender | | | | |
| Male | 371 | 58.3% | 344 | 58.1% |
| Female | 265 | 41.7% | 248 | 41.9% |
| IDH mutation | | | | |
| Yes | 336 | 52.8% | 372 | 62.8% |
| No | 300 | 47.2% | 220 | 37.2% |
| 1p19q codeletion | | | | |
| Yes | 126 | 19.8% | 149 | 25.2% |
| No | 510 | 80.2% | 443 | 74.8% |
| MGMTp methylation | | | | |
| Yes | 347 | 54.6% | | |
| No | 289 | 43.4% | | |
| Radio | | | | |
| Yes | 501 | 78.8% | | |
| No | 135 | 21.2% | | |
| Chemo | | | | |
| Yes | 465 | 73.1% | | |
| No | 171 | 26.9% | | |

Notes.
CGGA, Chinese Glioma Genome Atlas; TCGA, The Cancer Genome Atlas; IDH, Isocitrate Dehydrogenase; MGMT, O6-methylguanine-DNA methyltransferase.

*PLOD1* expression in glioma ranked 4th among the cell lines from different cancer tissues (Fig. 2H).

## Survival analysis and prognostic values of *PLOD1* in glioma patients

The Kaplan–Meier survival analysis of the CGGA datasets showed that high level expression of *PLOD1* related to poor survival ($p < 0.001$, Fig. 3A), and the same results were obtained by using the TCGA datasets ($p < 0.001$, Fig. 3B). Furthermore, we investigated the correlation between *PLOD1* expression and IDH and MGMT promoter methylation status on survival. The results showed that patients with IDH mutation had a longer survival regardless of the level of *PLOD1* expression (AUC = $p < 0.001$, Fig. 3C). However, patients with low expression of *PLOD1* had a longer survival regardless of the MGMT promoter methylation status ($p < 0.001$, Fig. 3D). In addition, based on the CGGA datasets, ROC curve revealed that *PLOD1* was a predictive marker of 1-year (AUC = 0.816), 3-year (AUC = 0.793), and 5-year survival (AUC = 0.707) (Fig. 3E). Similarly, ROC curve analysis using TCGA

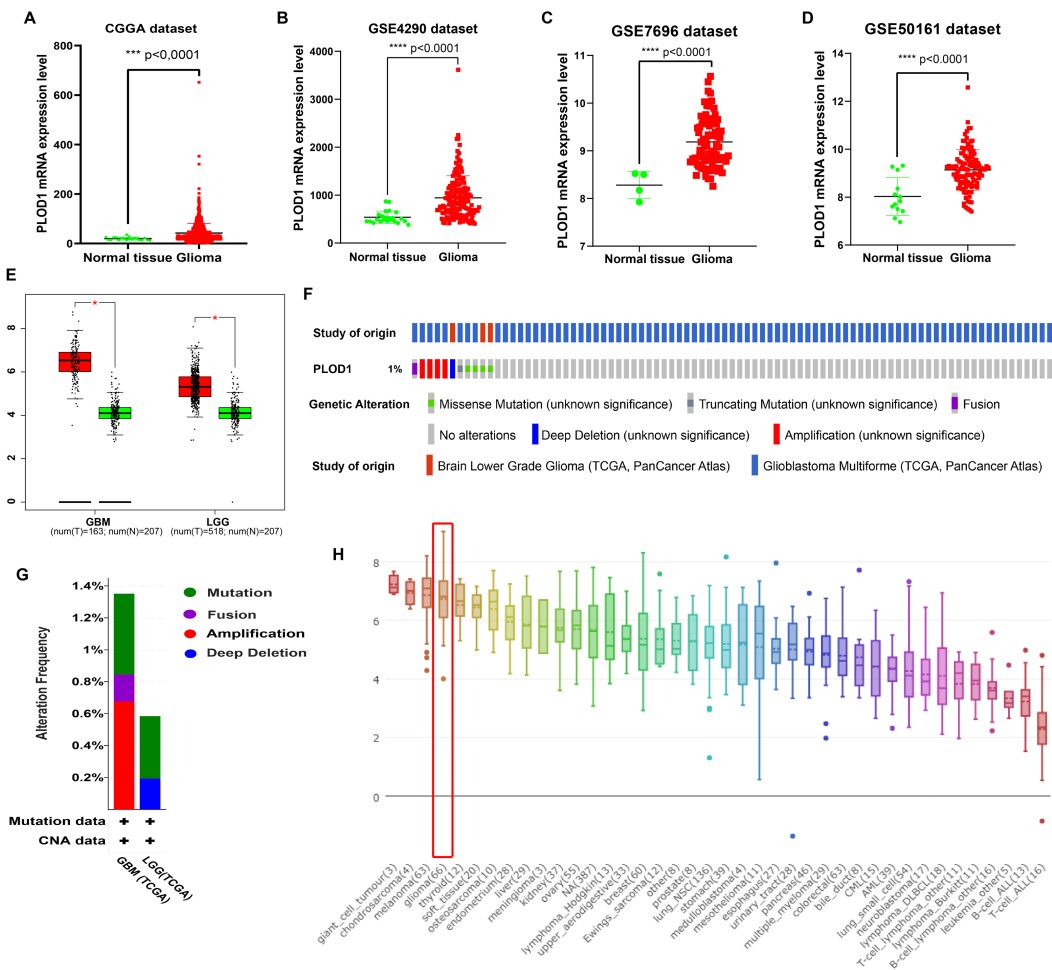

**Figure 2** **PLOD1 gene expression in glioma and normal tissue.** PLOD1 gene expression in glioma and normal tissue. (A) Comparison of PLOD1 expression levels between glioma and normal tissue in the CGGA datasets. (B) GSE4290 dataset. (C) GSE7696 dataset. (D) GSE50161 dataset. (E) PLOD1 expression levels of GBM and LGG in GEPIA analysis, comparing to normal tissue respectively. (F) The genomic alteration of PLOD1 using cBioPortal online analysis. (G) Mutation types of PLOD1. (H) PLOD1 expression level across various cancer cell lines, including glioma cell lines (rank 4th, indicated by red boxes) from the CCLE database (*Y*-axis represents the expression level of PLOD1 in different cancer cell lines).

datasets also verified this result. The area under curve for OS was 0.786 at 1 years, 0.805 at 3 years, and 0.796 at 5 years, respectively (Fig. 3F).

## Independent prognostic analysis and development of nomogram of *PLOD1* in glioma

To identify whether *PLOD1* was an independent prognostic index, univariate and multivariate Cox regression analyses were performed in CGGA datasets. Univariate analysis showed that *PLOD1* expression (HR = 1.986; 95% CI [1.796–2.198]; $P < 0.001$), PRS type, grade, age, IDH mutation, and 1p19q codeletion were significantly associated with OS (Fig. 4A). Furthermore, multivariate Cox regression analysis revealed that *PLOD1*

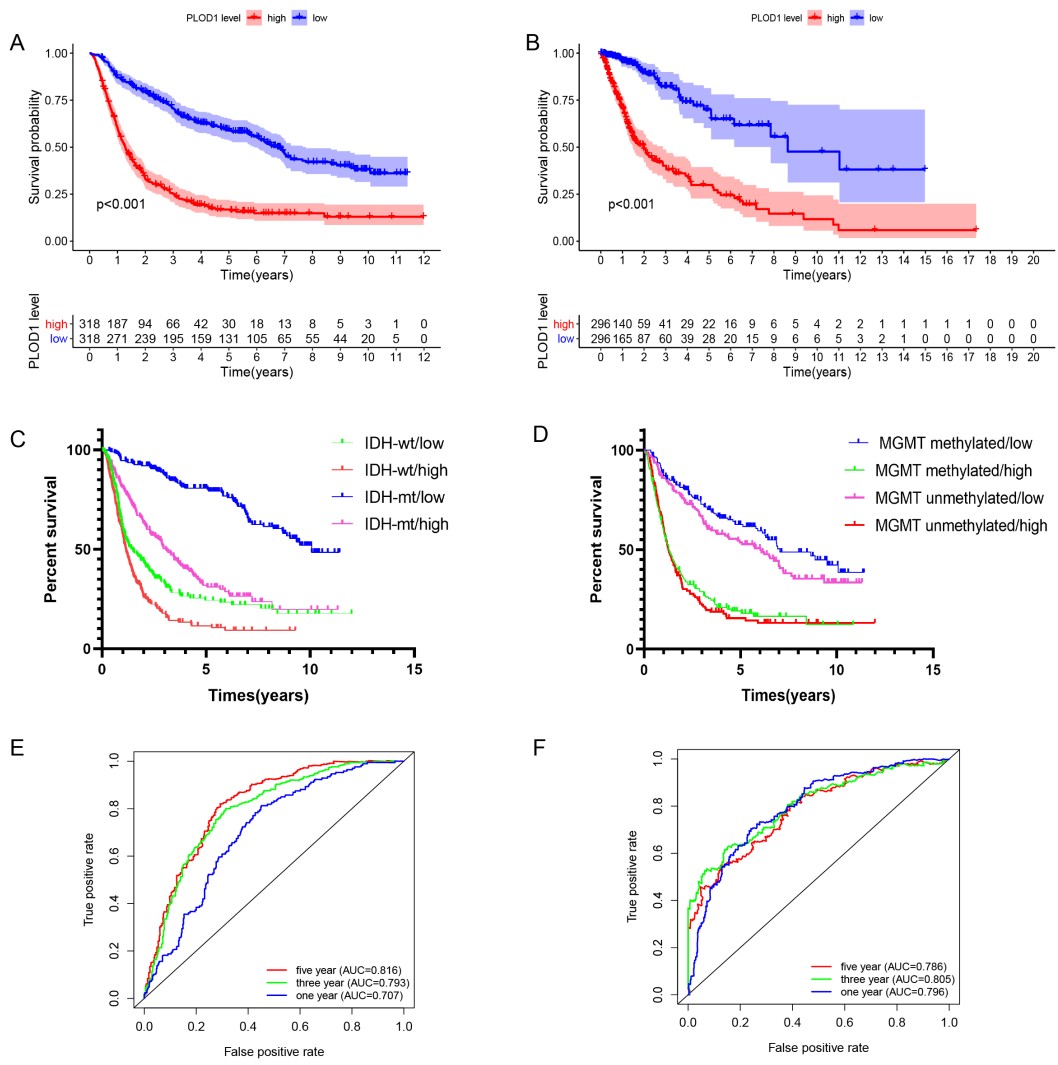

**Figure 3** **Survival analysis and prognostic values of PLOD1.** Survival analysis of glioma patients in the high PLOD1 and low PLOD1 groups. Red lines represent high expression and blue lines represent low expression. (A) Based on CGGA datasets. (B) Based on TCGA datasets. (C) Relationship between PLOD1 expression and IDH status on glioma survival. Green line represents low expression and IDH wildtype, red line represents high expression and IDH wildtype, blue line represents low expression and IDH mutation, purple line represents high expression and IDH mutation. (D) Relationship between PLOD1 expression and MGMT promoter methylation status on glioma survival. Blue line represents low expression and MGMT promoter methylation, green line represents high expression and MGMT promoter methylation, purple line represents low expression and MGMT promoter unmethylation, red line represents high expression and MGMT promoter unmethylation. Receiver operator characteristic curve analysis of PLOD1. Red lines represent 5 year survival, green lines represent 3 year survival and blue lines represent one 1 survival. (E) Based on CGGA datasets. (F) Based on TCGA datasets.

expression (HR = 1.283; 95% CI [1.128–1.460]; $P < 0.001$), PRS type, grade, chemotherapy after resection, IDH mutation, and 1p19q codeletion remained significantly correlated with OS (Fig. 4B). These results indicated that *PLOD1* expression has a strong prognostic value in gliomas. Furthermore, to quantitatively predict the prognosis of glioma patients, we

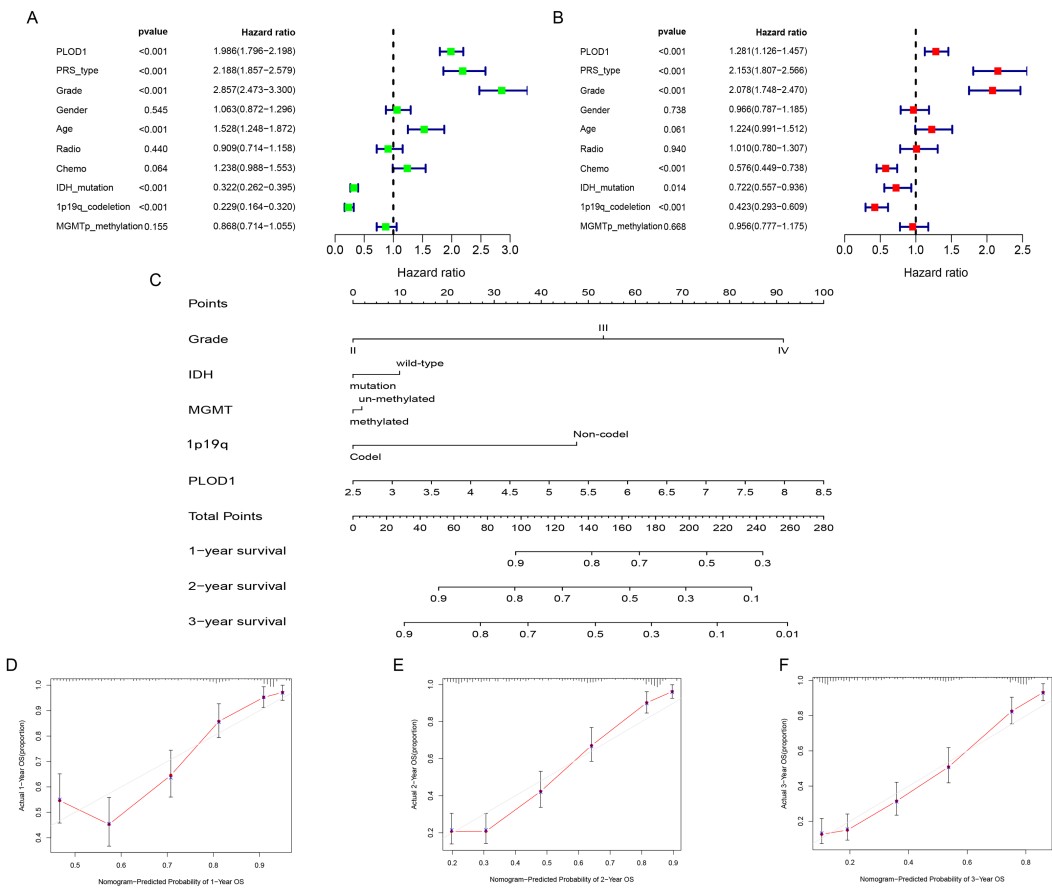

**Figure 4** **Independent prognostic analysis and development of nomogram of PLOD1 in glioma.** (A) Univariate analysis of PLOD1 based on CGGA datasets. (B) Multivariate analysis of PLOD1 based on CGGA datasets. (C) Prognostic nomogram to predict the survival of glioma patients based on the CGGA datasets. The values of grade, IDH, MGMT, 1p19q and PLOD1 are acquired from each variable axis. The total points on the axis are the sum values of these factors, which can predict the 1-, 2-, and 3-year survival. Calibration curves of the nomogram for predicting survival at 1, 2, and 3 years in the CGGA training cohort (D–F).

constructed a nomogram using grade, IDH mutation status, MGMT promoter methylation status, 1p19q codeletion status and *PLOD1* expression level. It revealed that the *PLOD1* expression level was the leading factor for predicting nomogram (Fig. 4C). Calibration curves indicated that actual and predicted survival matched very well, especially for 3-year survival (Fig. 4D–Fig. 4F).

## Correlations between *PLOD1* expression and clinicopathologic characteristics

Among the analysis, patients older than 42 years had significantly higher levels of *PLOD1* expression ($p < 0.001$, Fig. 5A). The expression of *PLOD1* increased with the increase of glioma grade ($p < 0.001$, Fig. 5B) and was higher in recurrent and secondary tumor than primary tumor ($p < 0.001$, Fig. 5C). For molecular type, *PLOD1* expressed lower in patients with 1p19q codeletion and IDH1 mutants ($p < 0.001$, Figs. 5D, 5E).

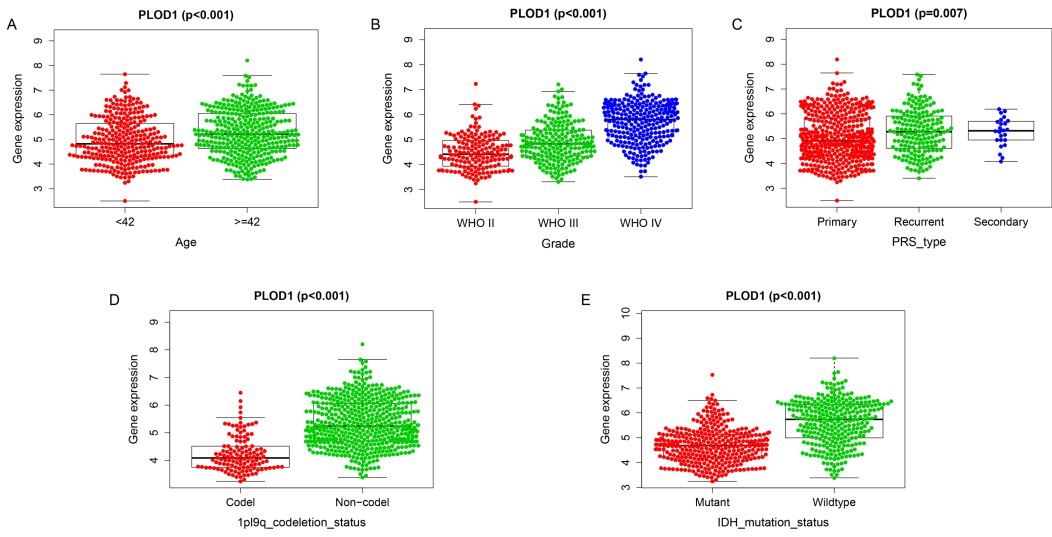

**Figure 5** **Correlation analysis with clinicopathologic characteristics of PLOD1 in glioma.** (A) Age ($<42$, $n = 294$; $\geq 42$, $n = 342$), (B) grade (WHO II, $n = 164$; WHO III, $n = 207$; WHO IV, $n = 265$), (C) PRS type (Primary, $n = 433$; Recurrent, $n = 179$; Secondary, $n = 24$), (D) 1p19q codeletion status (Codel, $n = 126$; Non-codel, $n = 510$), and (E) IDH mutation status (Mutant, $n = 347$; Wildtype, $n = 289$).

### DEGs and enrichment pathways analysis of *PLOD1*

A total of 67 DEGs were identified in analysis, which included 61 upregulated and 6 downregulated genes (Fig. 6A). Metascape analysis demonstrated that the biological processes of these DEGs were significantly enriched in extracellular structure organization (Fig. 6B). In addition, GSEA was used to identify hallmarks of glioma. The results showed that ECM receptor interaction was significantly enriched in high-expression *PLOD1* phenotypes (NES = 1.96, normalized $P = 0.002$) (Fig. 6C). As Metascape analysis and GSEA analysis both enriched in ECM related pathways, we screened out the core genes of ECM-related pathways enriched in GSEA analysis and intersected the core genes with DEGs to obtain 10 common genes (Fig. 6D).

### Co-expression analysis of *PLOD1*

Through gene co-expression analysis, we found that *PLOD1* was closely related to all of the screened common genes. The *PLOD1* was positively associated with *HSPG2*, *COL6A2*, *COL4A2*, *FN1*, *COL1A1*, *COL4A1*, *CD44*, *COL3A1*, *COL1A2*, *SPP1* (Figs. 7A–7J). A heatmap of the intersection genes associated with *PLOD1* was plotted (Fig. 7K), and a circular plot of these genes was also generated (Fig. 7L). Furthermore, we performed gene expression and survival analysis of the the common genes in GEPIA, the results showed that all the genes were highly expressed in glioma and high expression level related to poor prognosis (Figs. 8A–8T).

## DISCUSSION

Glioma, especially high-grade glioma, has a poor prognosis due to its aggressive growth and high recurrence rate. The standard treatment for glioma is a combination of

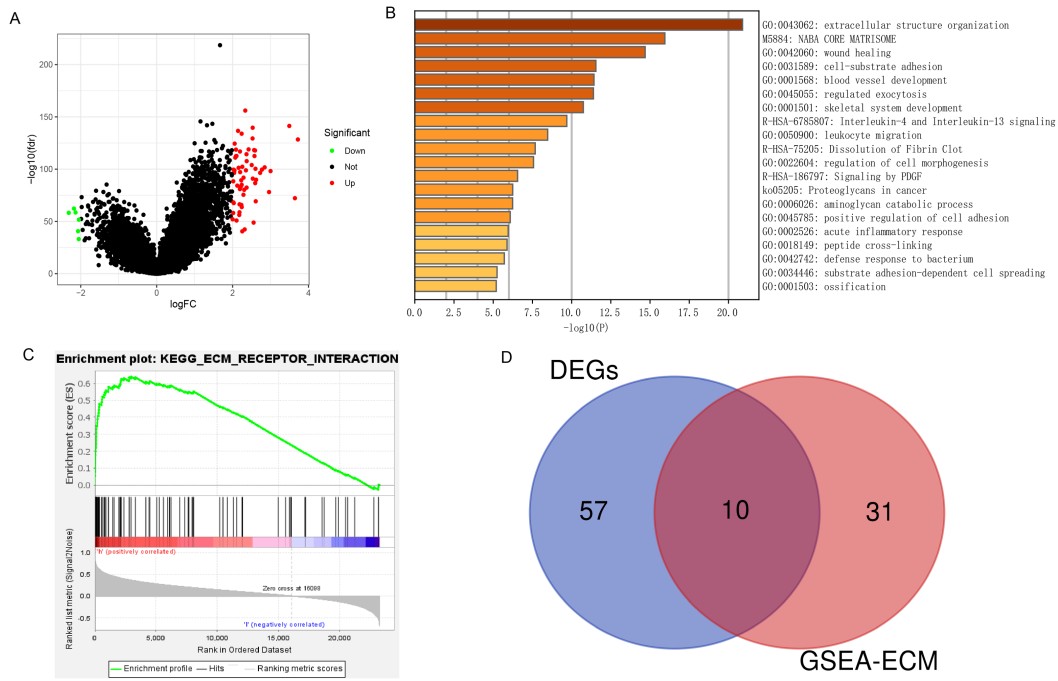

**Figure 6** **DEGs and enrichment pathways analysis of PLOD1.** (A) Volcano plot of differentially expressed genes. Red nodes represent the significantly up-regulated genes with logFC > 2 and $p < 0.05$. Green nodes represent the significantly down-regulated genes with logFC < −2 and $p < 0.05$. (B) Heatmap of enriched terms across DEGs in Metascape analysis. (C) Enrichment plot of ECM receptor interaction pathway from GSEA. (D) The Venn diagram of the common genes, which is obtained by intersecting the core genes of the ECM pathway with DEGs.

traditional surgery, radiotherapy and chemotherapy (*Buckner et al., 2016*; *Jaeckle et al., 2020*; *Herrlinger et al., 2019*). Emerging alternating electric field therapy has been included in the NCCN guidelines in recent years due to its significant survival benefits in glioblastoma (*Stupp et al., 2017*). However, the prognosis is still poor, and the choice of multiple treatment options is a clinical challenge. Therefore, it is important to explore potential biomarkers and therapeutic targets of glioma.

In this study, we found that *PLOD1* expression in glioma was higher than that in normal tissues using the CGGA datasets, which was verified by the GEO datasets and GEPIA. In addition, the expression of *PLOD1* in glioma was higher than in most cancers based on the CCLE database. Previous studies have revealed that *PLOD2* and *PLOD3* have predictive effects on the prognosis of glioma (*Song et al., 2017*; *Tsai et al., 2018*), we speculated that *PLOD1* also influence the prognosis of glioma, although it has not been explored. The sequential Kaplan–Meier survival analysis showed high expression level of *PLOD1* related to poor prognosis based on both the CGGA and TCGA datasets. We also found that patients with different IDH mutation and MGMT promoter methylation status had different survival at different *PLOD1* expression levels. Through comprehensive univariate and multivariate Cox analysis, we found that *PLOD1* was an independent prognostic factor in glioma patients. Moreover, the AUC values of ROC curves for *PLOD1* at 1, 3, and

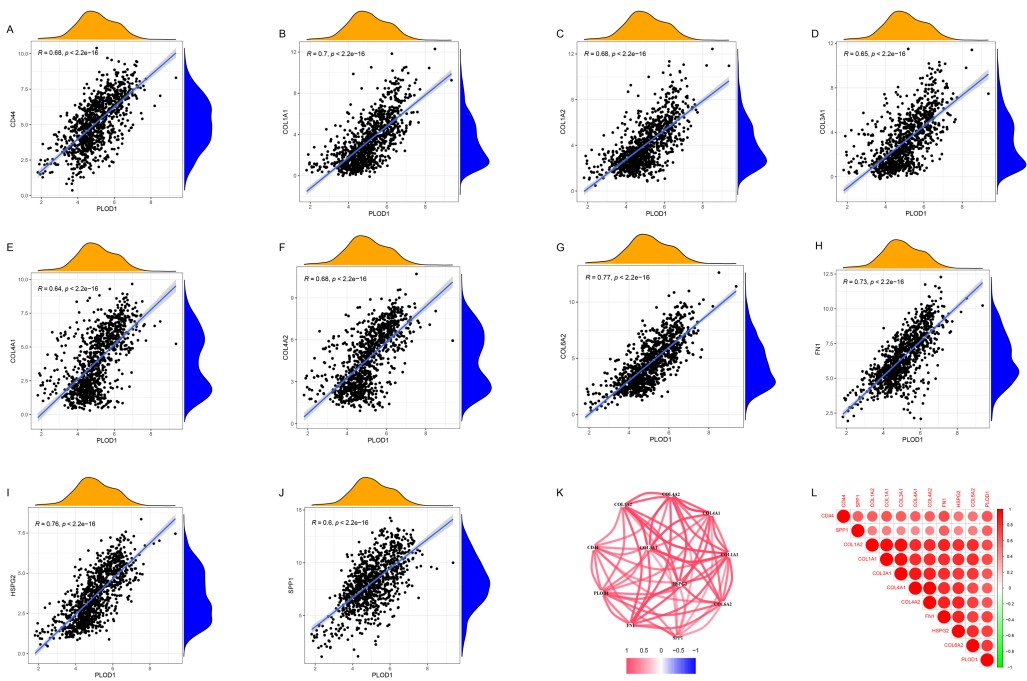

**Figure 7** **Co-expression analysis of PLOD1 with common genes.** (A) CD44; (B) COL1A1; (C) COL1A2; (D) COL3A1; (E) COL4A1; (F) COL4A2; (G) COL6A2; (H) FN1; (I) HSPG2; (J) SPP1. (K) Circular plot of common genes with PLOD1. Red represents positive association. (L) Heatmap of the common genes with PLOD1.

5 years were all >0.7, which also suggested that *PLOD1* was a predictor of survival. All of the above results confirmed our previous hypothesis. According to these prognostic signatures, we constructed a nomogram to quantitatively predict the survival of glioma patients, and the results showed that the *PLOD1* expression level was the leading factor. We can use this model to predict the survival of glioma patients. Therefore, it can help to make clinical decisions for patients, which can avoid overtreatment or undertreatment, so as to individually select the best treatment strategies for glioma patients.

ECM is an important component of tumor microenvironment and plays an important role in cancer development and progression (*Mohan, Das & Sagi, 2020*; *Lu, Weaver & Werb, 2012*). Collagen is a major component of ECM, and its elevated deposition and cross-linking can worsen tumor progression depending on the hydroxylation of lysine residues, which is mainly catalyzed by *PLODs* (*Gjaltema & Bank, 2017*; *Qi & Xu, 2018*). In this study, both GO and GSEA analysis were performed to suggest that *PLOD1* was enriched in ECM-related pathways, which was consistent with its pathophysiological mechanism. Using cBioPortal online analysis, we found that the mutation frequency of *PLOD1* is not high (1%), which suggests that the aberrant expression of *PLOD1* may be a result of post-transcriptional regulations or translation modifications. Previous studies reported that the *PLOD* expression was mainly regulated at the transcription level (*Gjaltema et al., 2015*). Compared with *PLOD2*, the regulation of *PLOD1* expression has not been well investigated. Some preliminary studies have shown that *PITX2* can directly regulate *PLOD1*

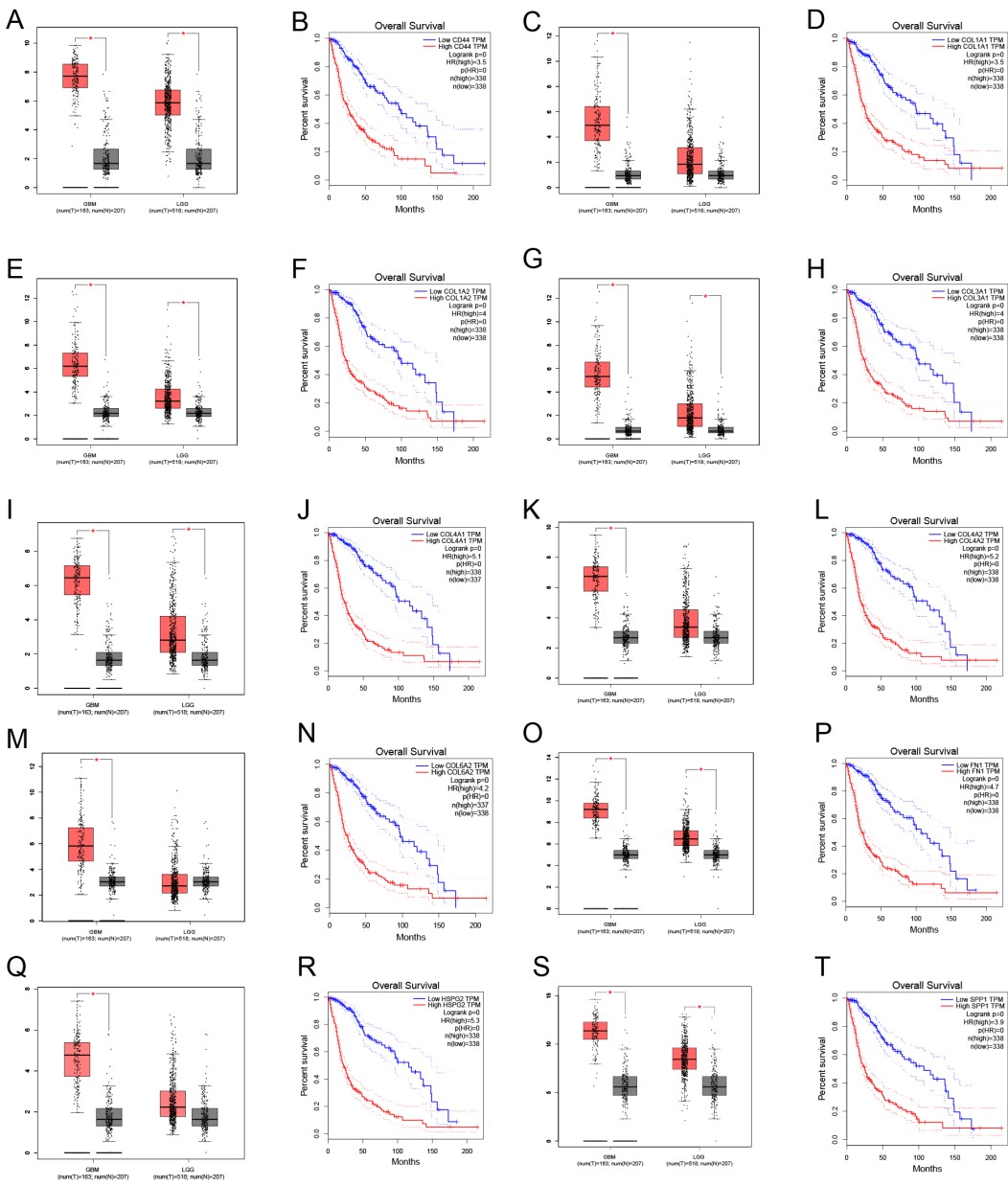

**Figure 8** **Common genes expression levels in glioma and correlations with survival in GEPIA analysis.** (A, B) CD44; (C, D) COL1A1; (E, F) COL1A2; (G,H) COL3A1; (I, J) COL4A1; (K, L) COL4A2; (M, N) COL6A2; (O, P) FN1; (Q, R) HSPG2; (S, T) SPP1.

expression by binding to the promoter region, using chromatin immunoprecipitation and luciferase reporting experiments (*Hjalt, Amendt & Murray, 2001*), and both *BMP-2* and *TGF- β1* can induce *PLOD1* expression in adipose tissue-derived mesenchymal stem cells (*Knippenberg et al., 2009*). However, the mechanism of its expression regulation still needs more exploration to identify.

As we found in glioma, increased expression of *PLOD1* is present in many types of cancer, and the high expression leads to short disease-related survival (*Wu et al., 2020*; *Yamada et al., 2019*; *Li et al., 2017*). Therefore, targeting *PLOD1* is a potential therapeutic strategy, while there is no potent *PLOD1* inhibitor available. So, it is of great significance to explore specific inhibitors of *PLOD1* for preventing tumor progression. In addition, another potential strategy is to reduce *PLOD1* expression, this means that further understanding of the regulatory mechanism of *PLOD1* in the development of cancer may lead to the exploration of novel signaling pathways to target *PLOD1*.

Finally, we screened out the common genes between the core genes of ECM-related pathways and DEGs, and then performed co-expression analysis with *PLOD1*. We found that *PLOD1* was positively related to *HSPG2, COL6A2, COL4A2, FN1, COL1A1, COL4A1, CD44, COL3A1, COL1A2, SPP1*, suggesting that they jointly contribute to the occurrence and development of gliomas. Moreover, based on GEPIA online analysis, all of them were highly expressed in glioma and its expression levels were closely related with patients' survival, therefore, it is of great significance to carry out more studies on these genes in glioma in the future.

## CONCLUSIONS

To analyze the relationship between the expression level of *PLOD1* and the prognosis of glioma, we use the CGGA, TCGA and GEO datasets performing bioinformatics analysis. The results showed that the expression level of *PLOD1* was higher in glioma than normal tissues and high expression of *PLOD1* was related to poor survival which can serve as an independent prognostic indicator for glioma patients. Additionally, the GO and GSEA analysis verified that the mechanism of *PLOD1*'s oncogenic effect was related to ECM, the co-expression analysis revealed that *PLOD1* was positively correlated with *HSPG2, COL6A2, COL4A2, FN1, COL1A1, COL4A1, CD44, COL3A1, COL1A2* and *SPP1*. All of them were ECM related genes and the expression of these genes was also correlated with the prognosis of glioma. In conclusion, this study indicated that targeting *PLOD1* is a potential therapeutic strategy for glioma patients, and the expression level of *PLOD1* may provide a reference for the selection of treatment regimens for glioma patients, which suggests that the biological functions and mechanisms of *PLOD1* need to be explored in the future. However, this study mainly relies on bioinformatics analysiswith certain limitations and more in vitro or in vivo validation experiments can make it more reliable.

## ACKNOWLEDGEMENTS

Thanks to all the researchers and staff working for The Cancer Genome Atlas database, The Cancer Genome Atlas database and Gene Expression Omnibus database.

### Funding

The authors received no funding for this work.

## Competing Interests

The authors declare there are no competing interests.

## Author Contributions

- Lei Tian, Huandi Zhou and Guohui Wang conceived and designed the experiments, performed the experiments, analyzed the data, prepared figures and/or tables, authored or reviewed drafts of the paper, and approved the final draft.
- Wen yan Wang analyzed the data, prepared figures and/or tables, and approved the final draft.
- Yuehong Li conceived and designed the experiments, authored or reviewed drafts of the paper, and approved the final draft.
- Xiaoying Xue conceived and designed the experiments, performed the experiments, prepared figures and/or tables, authored or reviewed drafts of the paper, and approved the final draft.

## Data Availability

The raw data are available in the Supplementary Files.

## Supplemental Information

Supplemental information for this article can be found online at http://dx.doi.org/10.7717/peerj.11422#supplemental-information.

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
