# Peer review of "The relationship between PLOD1 expression level and glioma prognosis investigated using public databases"

_PeerJ, doi:10.7717/peerj.11422_

## Round 0.1 · original submission · Minor Revisions

Please address all the concerns of the reviewers and amend your manuscript accordingly.

·

Basic reporting

The title, abstract, introduction, methods, results and discussion are appropriate for the content of the text. Furthermore, the article is well constructed, the experiments are well conducted, and analysis is well performed. The figures are relevant, high quality, well labelled and described.

Experimental design

The experimental design is original and the research is within the scope of the journal.. Research question is well defined, relevant and meaningful. The methods are highly technical, ethical and logistical.

Validity of the findings

All underlying data have been provided in detail. The findings are meaningful. The conclusions are well stated and relevant to the research questions.

Additional comments

This paper investigates the function of PLOD1 gene expression and mutation in glioma pathology and prognosis using CGGA, TCGA, GEO and CCLE datasets. The authors identified PLOD1 as a potential oncogene in glioma by comparing the expression level in glioma samples and normal controls. The authors demonstrate that PLOD1 is functionally related to ECM signaling pathways utilizing the GSEA method.

Editorial Criteria
BASIC REPORTING
The title, abstract, introduction, methods, results and discussion are appropriate for the content of the text. Furthermore, the article is well constructed, the experiments are well conducted, and analysis is well performed. The figures are relevant, high quality, well labelled and described.
EXPERIMENTAL DESIGN
The experimental design is original and the research is within the scope of the journal.. Research question is well defined, relevant and meaningful. The methods are highly technical, ethical and logistical.
VALIDITY OF THE FINDINGS
All underlying data have been provided in detail. The findings are meaningful. The conclusions are well stated and relevant to the research questions.

Overall, I think this paper is novel and will be of interest to the community of glioma genetics. The statistical part is valid and makes sense. The authors make it comprehensive by integrating analysis of multiple sources including CGGA, GEO, TCGA and CCLE. The main strengths of this paper is that it addresses an interesting and timely question, finds a novel solution based on a carefully selected set of rules and filters. As such this article represents an excellent and elegant bioinformatics study which will almost certainly influence our thinking about the function of PLOD1 in glioma. Some of the weaknesses are the lack of in vitro or in vivo validation experiments. In general, the work is convincing except some major and minor comments below:


Major Comments:

I’m wondering if there are any ongoing clinical trials focusing on PLOD1 in GBM or glioma patients? It will be very strong evidence for the significance of the current study if so.

I’m just wondering if GSE4290 is the only GEO dataset with glioma expression data? Given the sample size for GEO datasets are relatively small, I strongly recommend the authors to incorporate more data into their finding datasets. The authors may also check if there are some consortiums focusing on glioma.

To assess PLOD1 expression in different cancers, the authors used the CCLE data, which is a collection of cancer cell lines. I would recommend exploring the NCI's Genomic Data Commons (GDC) to see if you can find some evidence in the GBM and LGG patient samples in TCGA. Please focus on the “mutation” tab and “oncoGrid” tab.
(https://portal.gdc.cancer.gov/exploration?filters=%7B%22op%22%3A%22and%22%2C%22content%22%3A%5B%7B%22op%22%3A%22in%22%2C%22content%22%3A%7B%22field%22%3A%22genes.is_cancer_gene_census%22%2C%22value%22%3A%5B%22true%22%5D%7D%7D%5D%7D)



Minor Comments:
It is great that a session of abbreviations was there to list all the abbreviations for the database names. I would recommend to also include abbreviations like GBM, LGG, etc in that list.

All the gene names should be italic for all the gene names.

Please add the sample size of both glioma samples and controls for TCGA and GEO datasets in the Data collection and download part, although they were labelled in the flowchart. And for TCGA dataset, how many samples are LGG and how many for GBM? It’s better to make it clear in the Data collection as well.

Line 284: please include the lack of in vitro validation experiment as one of the limitations.

Figure2: the labels are difficult to read for Figure 2D, 2E, 2F. They were too small and not clear.

Reviewer 2 ·

Basic reporting

no comment

Experimental design

no comment

Validity of the findings

no comment

Additional comments

The paper addresses an important issue of relationship between PLOD1 expression level and glioma prognosis. It is well written, interesting and suggests a potential future therapeutic strategy for cancer patients. I think this article should be published.

---

## Round 0.2 · accepted · Accept

All concerns of the reviewers were adequately addressed and the revised version is acceptable now.